# Relationship between the Expression of Matrix Metalloproteinases and Their Tissue Inhibitors in Patients with Brain Tumors

**DOI:** 10.3390/ijms25052858

**Published:** 2024-03-01

**Authors:** Katarina Dibdiakova, Zuzana Majercikova, Tomas Galanda, Romana Richterova, Branislav Kolarovszki, Peter Racay, Jozef Hatok

**Affiliations:** 1Department of Medical Biochemistry, Jessenius Faculty of Medicine in Martin, Comenius University in Bratislava, Mala Hora 11161/4D, 03601 Martin, Slovakia; katarina.dibdiakova@uniba.sk (K.D.); majercikova.zuz@gmail.com (Z.M.); peter.racay@uniba.sk (P.R.); 2Department of Pathological Physiology, Jessenius Faculty of Medicine in Martin, Comenius University in Bratislava, Mala Hora 11161/4C, 03601 Martin, Slovakia; 3Department of Neurosurgery, Roosevelt Hospital, Slovak Medical University, Nam. L. Svobodu 1, 97517 Banska Bystrica, Slovakia; tgalanda@imafexbb.sk; 4Clinic of Neurosurgery, Jessenius Faculty of Medicine in Martin, University Hospital in Martin, Comenius University in Bratislava, Kollarova 2, 03601 Martin, Slovakia; richterova.romana@uniba.sk (R.R.); branislav.kolarovszki@uniba.sk (B.K.)

**Keywords:** brain tumors, metalloproteinases, tissue inhibitors, qRT-PCR, immunodetection

## Abstract

Matrix metalloproteinases (MMPs) and their inhibitors (TIMPs) play critical roles in regulating processes associated with malignant behavior. These endopeptidases selectively degrade components of the extracellular matrix (ECM), growth factors, and their receptors, contributing to cancer cell invasiveness and migratory characteristics by disrupting the basal membrane. However, the expression profile and role of various matrix metalloproteinases remain unclear, and only a few studies have focused on differences between diagnoses of brain tumors. Using quantitative real-time PCR analysis, we identified the expression pattern of ECM modulators (*n* = 10) in biopsies from glioblastoma (GBM; *n* = 20), astrocytoma (AST; *n* = 9), and meningioma (MNG; *n* = 19) patients. We found eight deregulated genes in the glioblastoma group compared to the benign meningioma group, with only *MMP9* (FC = 2.55; *p* = 0.09) and *TIMP4* (7.28; *p* < 0.0001) upregulated in an aggressive form. The most substantial positive change in fold regulation for all tumors was detected in *matrix metalloproteinase 2* (MNG = 30.9, AST = 4.28, and GBM = 4.12). Notably, we observed an influence of *TIMP1*, demonstrating a positive correlation with *MMP8*, *MMP9*, and *MMP10* in tumor samples. Subsequently, we examined the protein levels of the investigated MMPs (*n* = 7) and TIMPs (*n* = 3) via immunodetection. We confirmed elevated levels of MMPs and TIMPs in GBM patients compared to meningiomas and astrocytomas. Even when correlating glioblastomas versus astrocytomas, we showed a significantly increased level of MMP1, MMP3, MMP13, and TIMP1. The identified metalloproteases may play a key role in the process of gliomagenesis and may represent potential targets for personalized therapy. However, as we have not confirmed the relationship between mRNA expression and protein levels in individual samples, it is therefore natural that the regulation of metalloproteases will be subject to several factors.

## 1. Introduction

Glioblastoma (GBM), classified as World Health Organisation (WHO) grade IV, stands out as the most prevalent malignant glioma [1] and ranks among the most lethal human malignancies [2], with a median overall survival of merely 14 months from the time of diagnosis [3]. GBM can either progress from low-grade astrocytomas (such as diffuse astrocytoma, WHO II, or anaplastic astrocytoma, WHO III) or manifest without a preceding de novo clinical history [4]. Despite the implementation of supportive therapeutic strategies, the treatment paradigms for glioblastoma have seen limited evolution over the decades [5,6]. The established standard protocol involves surgical resection followed by radiotherapy and concomitant chemotherapy with temozolomide [7]. Regrettably, notwithstanding the application of multimodal treatment, the prognosis for patients diagnosed with glioblastoma remains bleak, with a 5-year survival rate ranging between 5% and 10% [8].

In contrast to glioblastomas, meningiomas arise as intracranial extra-axial neoplasia and, in the majority of cases, represent a benign form of central nervous system (CNS) tumors. Although some patients with completely resected grade I meningiomas are considered cured, there are cases with recurrence even after decades [9]. In addition to traditional surgical, radiotherapy, or hormonal approaches, immune therapy and targeted molecular approaches have come to the forefront in recent years [10]. Brain tumors, particularly glioblastoma, exhibit a high intratumor heterogeneity [11]. Key features include vascular disorganization, angiogenesis, and invasive characteristics [12]. Moreover, the rate of infiltration into the surrounding tissue is critical to cancer progression and recurrence [13]. Tumor cell invasion involves a complex interplay of several steps, including tumor cell migration and extracellular matrix disintegration. The disruption of the extracellular matrix (ECM) is crucial for tumor mass growth and cell invasion, creating space for malignant cells to migrate [14]. The matrix metalloproteinases (MMPs) and their inhibitors (TIMPs) play a pivotal role in this process, not limited to CNS tissue [15,16]. The MMP family comprises 23 endopeptidases that catalyze the degradation of the ECM and the basal membrane [17]. Increased MMP expression and activation also contribute to various pathological processes, such as rheumatoid arthritis, cardiovascular disease, and cancer progression [17,18]. Tissue inhibitors of metalloproteinases (TIMP1—TIMP4) are natural inhibitors of matrix metalloproteinases. Binding to active MMPs reversibly inhibits their proteolytic activity, maintaining homeostasis in the ECM [19].

The present study comparatively analyzed the MMP and TIMP family members based on transcriptomic and proteomic data in glioblastoma, astrocytoma, and meningioma biopsic samples. Simultaneously, we highlight the relationship between individual genes, specifically the correlation of selected MMPs with *TIMP1* or *TIMP2*. Our data indicate the significance of metalloproteinases and their inhibitors in the development of brain tumor processes.

## 2. Results

### 2.1. Absolute mRNA Level in Samples

To determine the absolute level of mRNA in brain samples, qRT-PCR analysis was conducted to characterize the transcriptomic profile of ten genes involved in extracellular matrix remodeling in glioblastomas, astrocytomas, and meningiomas. The RT^2^ PCR Array (Qiagen, Germantown, MD, USA) was employed to define the absolute transcriptomic pattern through mRNA expression. The mRNA expression of selected genes is presented as a logarithmic value of 2^−ΔCt^. Figure 1 illustrates the absolute values of these genes for all observed groups, including controls.

In the average of all products, the lowest amplification was observed in cases involving *MMP1*, *MMP3*, *MMP8*, and *MMP13* (Figure 1). Interestingly, the absence of mRNA expression of *MMP1* and *MMP13* was identified in both mRNA controls. Conversely, notably, the highest expression across all groups was recorded for the *TIMP2*, *TIMP1*, and *MMP2* genes. An elevated expression of *MMP2* in all testing groups, particularly in MNG, compared to the control group, was evident, with significant probability (2.2 × 10^−6^). In the case of the omitted gene, the meningioma group exhibited statistically significant differences compared to the remaining groups. Significant intergroup differences for *MMP8*, *MMP9*, and *MMP10* were noted among other metalloproteases (Figure 1). However, differences in the mRNA expression levels of *MMP1*, *MMP3*, and *MMP13* were not statistically significant in any test group when compared with the control and other test groups. Low levels of *MMP8* amplification were observed in all tumors, with astrocytomas showing a significant change compared to CTRL (*p* = 0.017) and MNG (*p* = 0.035). A statistically significant reduction in mRNA levels of *MMP10* in glioblastomas (*p* < 0.05) and astrocytomas (*p* = 0.022) compared to the control group was recorded.

A high mark of gene probability was achieved using Tukey’s post hoc analysis for all tissue inhibitors of MMP (Figure 1). Our analysis revealed statistically significant changes in *TIMP1* mRNA levels when comparing MNG (*p* < 0.001) and GBM (*p* < 0.001) with the control group, despite the presence of high mRNA levels of *TIMP1* in the control group. Significantly increased *TIMP1* levels were also detected in patients with astrocytomas and glioblastomas compared to patients with meningiomas (*p* < 0.001 and *p* = 0.024, respectively), and in patients with glioblastomas compared to those with astrocytomas (*p* < 0.001). For *TIMP2*, the only significant changes in mRNA levels were present in the astrocytoma and glioblastoma group when the meningioma group was taken into account (*p* < 0.001 for both changes). In meningiomas, the mRNA level of *TIMP4* was decreased compared to the glioblastoma (*p* < 0.001) and astrocytoma (*p* < 0.001) groups.

### 2.2. Differences in Gene Expression among Brain Tumors

Subsequently, we characterized the fold changes in the expression of MMP and TIMP genes in individual diagnoses when compared to the control group (Figure 2). Fold changes were calculated as the difference between 2^−ΔCt^ of a particular testing group and the control group and are presented as log_10_ 2^−ΔΔCt^. While the expression of *MMP2*, *MMP9*, *MMP10*, *TIMP1*, *TIMP2*, and *TIMP4* was observed in all samples within the screened diagnoses, *MMP3* expression was present in 88.2%, and *MMP8* in 94.1% of cases, respectively. As the genes for *MMP1* and *MMP13* were not amplified in the control samples (Figure 2), it was not possible to determine the fold change in these genes in the examined brain tumors. Moreover, in the case of *MMP2*, there was only one value for the control group, making it impossible to evaluate a statistical correlation. Despite this, we assert that the most substantial positive change in fold regulation for all genes was detected in tumor groups, precisely in matrix metalloproteinase 2 (Figure 2a; MNG = 30.9, AST = 4.28, and GBM = 4.12). Another upregulation compared to the control was detected only in the glioblastoma group for the *MMP9* (2.43) and *TIMP1* (10.3) genes, both without significance, as well as in meningiomas (*TIMP1* = 22.4). The remaining genes were downregulated in all tumor groups, with the most pronounced changes observed for *MMP8*, *MMP3*, and *MMP10* (Figure 2a). Among the tissue inhibitors of metalloproteinases, we determined a statistically significant downregulation of the gene *TIMP4* (−11.2) in meningiomas and *TIMP2* (−2.0) in astrocytomas.

Overall, changes in fold regulation with statistical significance were recorded for *MMP3* (*p* < 0.001), *MMP10* (*p* < 0.001), and *TIMP2* (*p* < 0.05) in glioblastoma. Similar results were obtained in astrocytomas: specifically, in the genes *MMP3*, *MMP8*, *MMP10*, *TIMP2*, and *TIMP4*. Significant fold changes were found in *MMP10*, *TIMP1*, and *TIMP4* in meningioma patients (Figure 2a).

In our exploration of regulation correlation, we turned our attention to understanding the relationship between malignant and benign samples. It proved interesting that the majority of monitored genes exhibited downregulation in the malignant group (Figure 2b). Significant downregulation was observed for *MMP2*, *MMP8*, *TIMP1*, and *TIMP2*. Conversely, the upregulation of the *TIMP4* gene was typical for malignant forms, including glioblastomas and astrocytomas with grades III to IV (Figure 2b). Additionally, we noted a positive change of more than twofold in *MMP9*, although without statistical significance.

### 2.3. Linear Correlation of mRNA Expression between Matrix Metalloproteinases and Their Inhibitors

To investigate the correlation between the expression of MMPs and TIMPs, we conducted a linear correlation analysis using Pearson’s r with OriginLab Pro 8.5 software (Figure 3). The only statistically significant correlations that we observed were in the correlation of *TIMP1* versus *MMP8*, *MMP9*, and *MMP10* (Figure 3a–c). *MMP10* even exhibited a positive correlation with *TIMP2* (*p* = 0.0141, Figure 3d). The remaining correlations between mRNA matrix metalloproteases and tissue inhibitors of MMPs showed a negative trend, although without statistical significance.

### 2.4. Absolute Protein Levels in Tissue Samples

To assess the protein levels of MMPs and TIMPs in our samples, a dot blot analysis was conducted. A positive signal was represented by a dot on a membrane (Figure 4), as provided in a dot blot array kit. All positive signals were normalized to internal controls present on the individual membranes, and chemiluminescent signal intensities were compared to those of a healthy human brain control (Figure 4).

An elevation in the protein levels of MMP1, MMP2, MMP3, and MMP8 was observed in astrocytoma samples, whereas a decrease in the amount of these proteins was noted in the glioblastoma group compared to the control group. An increase in the protein levels of MMP1, MMP2, MMP3, and MMP8 was observed in astrocytoma samples, while a reverse trend was noted, with a decrease in the amount of these proteins in the glioblastoma group when compared to the control group (Figure 5). In meningiomas, the protein levels of MMP1, MMP2, and MMP3 were similar to controls, but a slight decrease in MMP8 protein levels was identified. However, these changes were not statistically significant. The protein expression of MMP9, MMP13, TIMP1, TIMP2, and TIMP4 was elevated in all three investigated brain tumor diagnoses. Notably, patients with glioblastomas exhibited almost consistently elevated protein levels compared to the other groups, except for MMP10 and TIMP2 (Figure 5).

## 3. Discussion

Highly invasive characteristics, increasing the risk of disease recurrence, are common features of malignant gliomas and therefore remain a significant medical challenge. The present study evaluated the expression of genes involved in extracellular matrix remodeling in patients with confirmed astrocytomas, glioblastomas, or meningiomas. The main aim was to determine the expression profile of MMPs and TIMPs associated with brain tumors and characterize differences between diagnoses. The production of MMPs in the central nervous system under normal conditions is low; however, it can be augmented through neurodegenerative and neuroinflammatory pathologies, as well as through tumorigenic processes [20,21]. The deregulation of MMP expression is associated with the malignant phenotype and has been observed in gliomas [22,23], where an increase in the expression of various MMPs is correlated with a worse prognosis [24].

Béliveau et al. analyzed the expression of *MMP2*, *MMP9*, and *MMP12* in 60 brain tumor samples (WHO I-IV grade), including glioblastomas and meningiomas, among others [25]. They observed elevated expression of *MMP9* and *MMP2* in anaplastic astrocytomas, oligodendrogliomas, and glioblastomas compared to meningiomas and other benign tumors, indicating a direct relation between *MMP9* and *MMP2* expression and malignant behavior. However, it is important to highlight that their study compared eight different diagnoses via Western blot analysis, with no correlation to healthy tissue. We have previously highlighted, in a previous publication [26], the impact of proper control selection on the interpretation of results. Therefore, we decided to use commercially available samples from healthy brain tissue. Contrary to their study, our analysis revealed overexpression only in the *MMP9* gene exclusively in GBM compared to astrocytomas. Of interest was our finding of high *MMP2* expression in the meningioma group among all brain samples. Compared to the control sample, we also observed higher levels of *MMP2* in GBM and AST, but without statistical significance. Additionally, the *MMP1* and *MMP13* genes were undetectable in healthy brain controls. Therefore, we can conclude that they are overexpressed in all tumor groups, especially in glioblastomas, which was confirmed via dot blot analysis.

Different studies suggest that high MMP8 protein levels might predict better survival in tongue cancer patients [27] and some breast cancer patients [28] but provide a worse prognosis in hepatocellular and ovarian cancers [29,30]. The role of MMP8 in brain tumors is poorly understood to date. The expression level of MMP8 was reduced in all the groups that we monitored, even statistically significantly in MNG and AST. However, at the protein level, we obtained a similar trend compared to the control, except for glioblastomas, where we observed a nearly 1.5-fold increase. Its expression does not appear to be important for the tumorigenesis of brain tumors, which is supported by the work of Chernov, where MMP8 expression was not detected in multiple brain cell lines [31]. However, the role of post-transcriptional regulation remains questionable in terms of negative correlation, as we have shown elevated levels of MMP8 protein in GBM.

MMP3 belongs, together with MMP10, to the stromelysin subfamily [32]. In all monitored groups, we observed a decrease in their expression, which, in some groups, was of statistical significance. A biostatistical analysis of data from 276 histologically confirmed gliomas found that a high level of MMP3 expression and a low level of MMP10 were directly related to worse survival [33]. In our case, we unequivocally confirmed the negative expression of both MMPs at the gene level, which, in the case of *MMP10*, was also of significance for the group of astrocytomas and glioblastomas. On the other hand, via dot blot analysis, we noted increased levels of MMP3 and MMP10 in the GBM group. Likewise, increased levels of MMP3 were also identified by Mercapide et al. Their experiments confirmed that the secretion of metalloprotease 3 correlates with the tumorigenicity and invasive nature of astrocytomas and glioblastomas [34].

The balance between matrix metalloproteinases and their inhibitors is crucial for controlling glioma cell invasion and tumor growth [25,35]. To date, four TIMPs (TIMP1, -2, -3, and -4), expressed by a variety of cell types and present in most tissues and body fluids, have been characterized in humans [36]. Although they all inhibit MMPs’ proteolytic activity, TIMPs differ in many aspects, including solubility, interaction with the proenzymes (proMMPs), and regulation of expression. TIMP1, TIMP2, and TIMP4 are present in soluble forms, while TIMP3 is tightly bound to the matrix [21]. We observed the most significant differences both at the transcriptomic level and in the protein analysis of the metalloprotease inhibitors.

High TIMP1 expression is strongly associated with a poor prognosis in almost all known cancer types, which corresponds with the observed increase in this marker at both the transcriptomic and proteomic levels, especially in the GBM group. On the other hand, TIMP1 deficiency is associated with increased sensitivity [37] or its overexpression is connected with resistance to chemotherapy in some tumors [38]. A comprehensive analysis of the four TIMPs identified by Han’s collective suggests low *TIMP1* expression in MGMT-methylated patients and thus a better prognosis in patients with downregulation of this gene [39]. The transcriptomic analysis of the Groft scientific group showed a positive correlation between *TIMP1* expression and a negative correlation of *TIMP4* expression with tumor grade and malignant behavior, while the expression of *TIMP3* and *TIMP2* remained unchanged [40]. However, contrary to these results, the expression analysis of several genes, including *TIMP4*, showed its overexpression [41]. Our results support findings obtained by Yin et al. [41]—the expression of the gene encoding the tissue inhibitor of metalloproteinase 4 was increased by more than 13 times in the glioblastoma group compared to the meningioma group, confirming that *TIMP4* expression is positively correlated with tumor grade.

Lu et al. focused on the effect of TIMP2 deregulation on the glioblastoma line U87. They observed that the exogenous addition of comparable levels of purified TIMP2 to parental U87MG cells increased MMP2 activation and invasion, but the higher amounts of TIMP2 resulted in the inhibition of MMP2 activation, indicating that the complex balance between TIMP2 and MMP2 is a critical determinant of glioblastoma invasion [42]. Our proteomic analysis suggests a correlation between increased TIMP2 and MMP2 protein levels in GBM biopsy, but without statistical significance. At the mRNA level, there are significant differences, especially in individual diagnoses.

When *TIMP2* is taken into account, several authors have shown no correlation between *TIMP2* expression and tumor grade [35,43], while others have detected reduced mRNA levels of *TIMP2* in glioblastomas [44]. Determining *TIMP1* and *TIMP2* mRNA levels confirmed that the correlation with tumor grade is negative. The expression of both inhibitors was decreased exclusively in higher-grade malignancies (II and III) but was significantly higher in low-grade tumors [25]. Consistent with previous results, we confirmed a higher expression of *TIMP1* and *TIMP2* in the benign meningioma group. Among tissue MMP inhibitors, *TIMP4* appeared to be the only one overexpressed in malignant forms of gliomas. A more-than-twofold upregulation was observed for *MMP9*, but without statistical significance, in the same group of high-grade samples. Although the *MMP9* gene alone is considered a prognostic marker for glioblastomas, we are the only ones to point to the role of the malignancy grade in relation to *TIMP4*. However, when analyzing all samples, we clearly showed a relationship of *MMP9* with *TIMP1*, similar to Pietrzak et al. in their correlation with the survival of NSCLC patients [45]. Moreover, at the gene level, we also statistically demonstrated a positive correlation of *TIMP1* with *MMP8* and *MMP10*. The mRNA of *TIMP2* demonstrated its positive relationship with *MMP10*, despite the fact that *MMP10* expression was at lower amplifications in the patient samples, which did not correlate with our protein analysis. The dual nature of TIMP2 action in different tumor types has already been pointed out by Kaczorowska et al. in an earlier publication, which only confirms the presence of an unclear mechanism of action [46]. They summarized that the essence of the role of TIMP2 in cancer development is based on the protein’s concentration within individual tissues. A low concentration of TIMP2 activates matrix metalloproteinase 2, whereas a high concentration inhibits its activity. This phenomenon is attributed to the origin of malignant tissue, as well as various factors (stage of diseases and tumor size) or regulatory molecules (topoisomerase II, p53, and bcl-2) [46].

Of interest was the discordant correspondence between gene expression and protein levels. In the GBM samples, we obtained increased signal intensities at the protein level for almost all measured products, which was rather exceptional in the case of genes. Therefore, further experiments are necessary to confirm the mechanism associated with the post-translational modifications of products.

Matrix metalloproteinases play an essential role in the tumorigenesis of various tumors, including brain tumors. Their impact on cell growth, apoptosis, and inflammatory processes is one of many hallmarks of cancer enhancing the tumorigenicity and invasiveness of cancer cells. Explaining these mechanisms could yield exciting results in this area and could also open up new potential therapeutic options for the treatment of aggressive brain tumors.

## 4. Material and Methods

### 4.1. Tissue Samples

Tissue samples were obtained from a cohort of 51 patients diagnosed with brain tumors. Among these patients, 20 were diagnosed with glioblastomas (the median age at diagnosis was 57 years), 19 with meningiomas (64.5 years), and 9 with astrocytomas (48 years). In addition, we divided the patient samples into two groups: benign and malignant. The benign group consisted of meningioma patients (*n* = 19) and two astrocytoma patients with a maximum tumor grade of II. In the malignant group, glioblastoma patients (*n* = 20) with grade IV predominated, followed by seven cases of grade III astrocytomas. Surgical tumor resection was performed on all patients either at the Clinic of Neurosurgery (University Hospital in Martin) or at the Department of Neurosurgery (Roosevelt Hospital in Banska Bystrica) between 2018 and 2022. Immediately following surgery, tissue specimens were placed in RNAlater solution and subsequently frozen at −80 °C. All patients were informed of the study and provided informed consent prior to participation.

### 4.2. Control Samples

For real-time analysis, two commercially available human RNAs were used as control samples. Both represented total RNA from healthy brain tissue sourced from different companies: (i) Human Brain Total RNA (HR-201, Amsbio—Abingdon, UK) and (ii) Human Brain Total RNA (636530, Takara—Saint-Germain-en-Laye, France).

For dot blot analysis, commercially available lysates of total proteins were utilized as control samples: (i) Human Adult Normal Tissue—Brain (P1234035, Amsbio—Abingdon, UK) and (ii) Human Adult Normal Tissue—Brain Cerebral Cortex (P1234042, Amsbio—Abingdon, UK).

### 4.3. RNA Extraction

Total RNA was extracted from 20–30 mg of tissue specimens following the manufacturer’s protocol using the AllPrep^®^ DNA/RNA Mini Kit (Qiagen Inc., Germantown, MD, USA). An optional DNase I treatment step was performed during RNA extraction to eliminate DNA contamination in the samples. Subsequently, RNA concentration and purity were determined in 30 μL of RNase-free water by measuring absorbance at 260/280 nm. RNA integrity was assessed using a microchip electrophoresis system MultiNA—Shimadzu, Kyoto, Japan.

### 4.4. Quantitative RT-PCR Analysis

Equal amounts of isolated RNA (2 μg) were reverse-transcribed into cDNA using the RT^2^ First Strand Kit (Qiagen Inc., Germantown, MD, USA) with random hexamers and oligo-dT primers, which includes a genomic DNA elimination step. The obtained cDNA was diluted to a total volume of 111 μL with nuclease-free water.

Expression analysis of selected MMPs and TIMPs at the mRNA level was conducted via quantitative RT-PCR using a custom RT^2^ PCR Array (Qiagen Inc., Germantown, MD, USA) in a 96-well plate format (refer to Table 1). PCR samples were prepared to a final volume of 464 μL, consisting of 224 μL of SYBR^®^ Green ROX^TM^ qPCR Mastermix (Qiagen Inc., Germantown, MD, USA) and 224 μL of nuclease-free water. The reaction volume in each well was 25 μL.

Amplification was completed according to the manufacturer’s recommendations in ViiA7 PCR System (Life Technologies, Carlsbad, CA, USA). Three endogenous control genes (housekeeping genes)—*GAPDH* (glyceraldehyde-3-phosphate dehydrogenase), *ACTB* (β-actin), and *B2M* (β2-microglobulin) present on the PCR array were used for normalization. Amplification of cDNA was initiated via denaturation at 95 °C for 10 min followed by 40 PCR stage (95 °C, 15 min; 60 °C, 1 min) and melting curve stage (95 °C, 15 min; 60 °C, 1 min; 95 °C, 15 min). The relative expression of all genes in biopsic samples was calculated using the RT^2^ Profiler PCR Array Data Analysis Web Portal (Qiagen) based on 2^−ΔΔCt^ method [47], where ΔΔCt = (Ct_GOI_ − Ct_HKG_)_TEST GROUP_ − (Ct_GOI_ − Ct_HKG_)_CONTROL GROUP_. Fold change calculations were performed using Qiagen GeneGlobe—RT^2^ Profiler PCR Data Analysis (https://dataanalysis2.qiagen.com/pcr, accessed on 1 January 2023). The genes with a significant difference in expression were those with an average fold change of ≤−2.0 or ≥2.0, and statistically significant differences were those with a corresponding *p* value of <0.05.

### 4.5. Total Protein Isolation

For proteomic analysis, a subset of samples (*n* = 18) previously analyzed at the transcriptomic level was selected. Biopsic tissue (~10 mg) obtained from glioblastomas (*n* = 7), astrocytomas (*n* = 5), and meningiomas (*n* = 6) was thoroughly washed in 1× DPBS solution, dried, and subsequently homogenized using a SONOPLUS HD 3100 ultrasonic homogenizer (Bandelin, Berlin, Germany) in 500 µL of homogenization solution for 25 s. Following homogenization, the sample was centrifuged for 15 min at 4 °C at 12,000× *g*. The supernatant obtained was used to determine the protein concentration using the DC Protein Assay kit (Bio-Rad Laboratories—Hercules, CA, USA) and used for subsequent analyses.

### 4.6. Dot Blot Analysis

The presence of matrix metalloproteinases (MMPs) and their inhibitors (TIMPs) in protein samples was determined using the Dot Blot Array kit (Abcam, Cambridge, UK). Antibodies against MMPs and TIMPs were captured in duplicate as dots on the membrane. One membrane was coated with 1 mL of the prepared protein sample at a concentration of 250 μg/mL. As a control sample, total brain protein lysate was utilized (Amsbio, Cambridge, MA, USA).

Following incubation with the sample, biotinylated antibodies and HRP-conjugated streptavidin were bound to the immobilized antibodies. The procedure was carried out in accordance with the manufacturer’s instructions. Subsequently, detection was performed using a Molecular Imager ChemiDocTM XRS+ imaging system (BioRad, Hercules, CA, USA). The obtained signals were analyzed using Image Lab software, version 6.0.1 (Bio-Rad Laboratories—Hercules, CA, USA).

### 4.7. Statistical Analysis

The relative mRNA levels of MMPs and TIMPs were compared to control mRNA via Welch’s ANOVA and Tukey’s HSD post hoc test. Gene expression and fold changes in gene expression were evaluated using online analyzer RT^2^ Profiler PCR Data Analysis (https://dataanalysis2.qiagen.com/pcr, accessed on 1 January 2023; Qiagen). The significance of differences in gene and protein expression was determined using Student’s *t*-test (*p* < 0.05 *; *p* < 0.01 **; *p* < 0.001 ***). Fold change in gene expression >2.0 or <−2.0 with *p*-value < 0.05 was considered statistically significant.

Linear regression analysis was used to determine the relationship between TIMP and MMP RNA expression dates using OriginPro 8.5 SR1 software (OriginLab Corporation, Northampton, MA, USA). The strength of this relationship was defined by using the R value (Pearson’s correlation coefficient). We included only significant correlations in the results (*p* ≤ 0.05).

## 5. Conclusions

Proteolytic enzymes belonging to the catalytic group known as metalloproteinases play pivotal roles in various biological processes across practically all organisms. They can function as intracellular or extracellular regulators. Matrix metalloproteinases actively participate in tissue remodeling under both physiological and pathological conditions by degrading major protein components of the extracellular matrix. It is well established that they are often deregulated in many types of human cancers, and their increased activity can promote cancer progression by enhancing cancer cell growth, migration, invasion, metastasis, and angiogenesis. Given the high heterogeneity observed in brain tumors, particularly in glioblastoma multiforme (GBM), not only between individual patients but also within the tumor itself, the findings regarding metalloproteinases or their inhibitors are notably inconsistent. These discrepancies underscore the importance of MMPs as potential targets for research aimed at developing personalized therapies for brain tumors.

We observed an increase in *MMP2* mRNA levels in meningiomas compared to healthy samples, where *MMP1* and *MMP13* expression was not amplified in the control group. A significant increase was also noted in *TIMP1* expression in both MNG and GBM patients. The tissue inhibitor of metalloproteinases 1 was found to be crucial in correlating with selected metalloproteinases; we confirmed its positive relationship with *MMP8*, *MMP9*, and *MMP10*. When comparing the fold change expression data between the malignant and benign groups, we observed a predominance of downregulated genes, contrasting with the upregulation seen in *MMP9* and *TIMP4*. Interestingly, at the protein level, we confirmed elevated levels in GBM patients for almost all the studied markers, notably MMP1, MMP9, MMP13, TIMP1, and TIMP4.

## Figures and Tables

**Figure 1 ijms-25-02858-f001:**
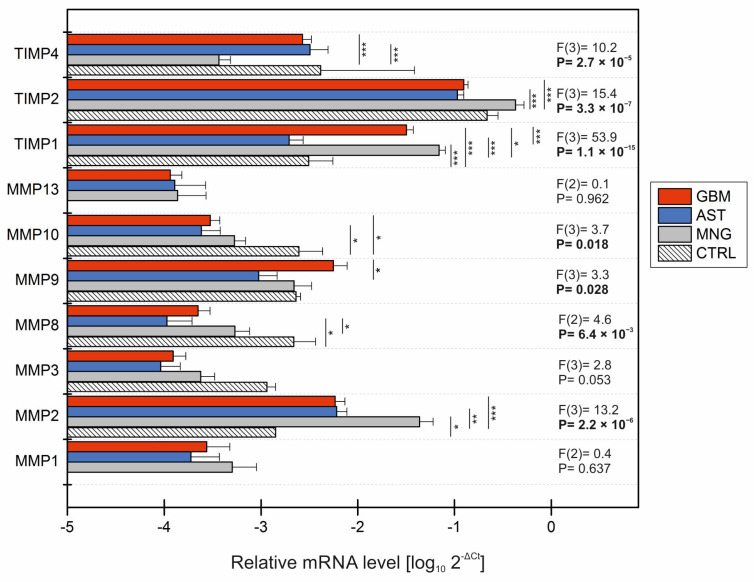
Relative logarithmic mRNA levels of MMPs and TIMPs. Each column represents the absolute mRNA level of an individual gene in the glioblastoma (GBM, red), astrocytoma (AST, blue), meningioma (MNG, grey), and control (CTRL, diagonal pattern) group, where each subgroup stands for a particular gene. The F value (factor score) was calculated for all samples in the dataset using one-way ANOVA and Tukey’s post hoc test. To assess statistically significant changes between particular groups, Student’s *t*-test was used. * *p* < 0.05; ** *p* < 0.01; *** *p* < 0.001. All mRNA levels were normalized to the levels of endogenous controls. Transcriptomic data are represented as a logarithmic of 2^−ΔCt^ + SEM. MMP—matrix metalloproteinase; TIMP—tissue inhibitor of metalloproteinases.

**Figure 2 ijms-25-02858-f002:**
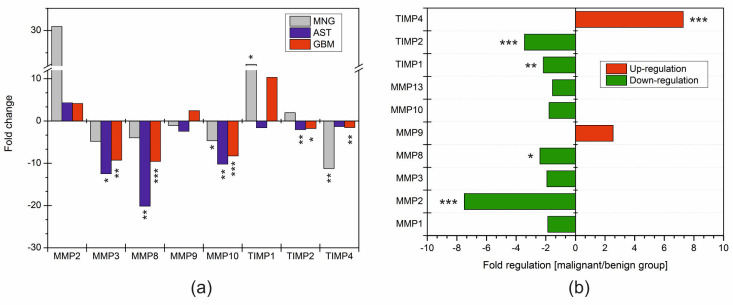
Comparison of logarithm fold change ratios for MMP and TIMP genes in brain tumors. (**a**) The individual column represents the log_10_ 2^−ΔΔCt^ values of meningiomas (MNGs, grey), astrocytomas (ASTs, blue) and glioblastomas (GBMs, red) compared to the control group. Fold change values greater than two indicate upregulation and values less than two indicate downregulation. *p*-values were obtained with a Student’s *t*-test; * *p* < 0.05, ** *p* < 0.01, *** *p* < 0.001. (**b**) The fold regulation of MMP and TIMP genes between malignant (*n* = 27) and benign (*n* = 21) samples. The green bars represent the downregulated genes in the malignant group compared to the benign group. In contrast, the red bars represent upregulated genes. *p*-values were obtained with a Student’s *t*-test; * *p* < 0.05, ** *p* < 0.01, *** *p* < 0.001. MMP—matrix metalloproteinase, TIMP—tissue inhibitor of metalloproteinases.

**Figure 3 ijms-25-02858-f003:**
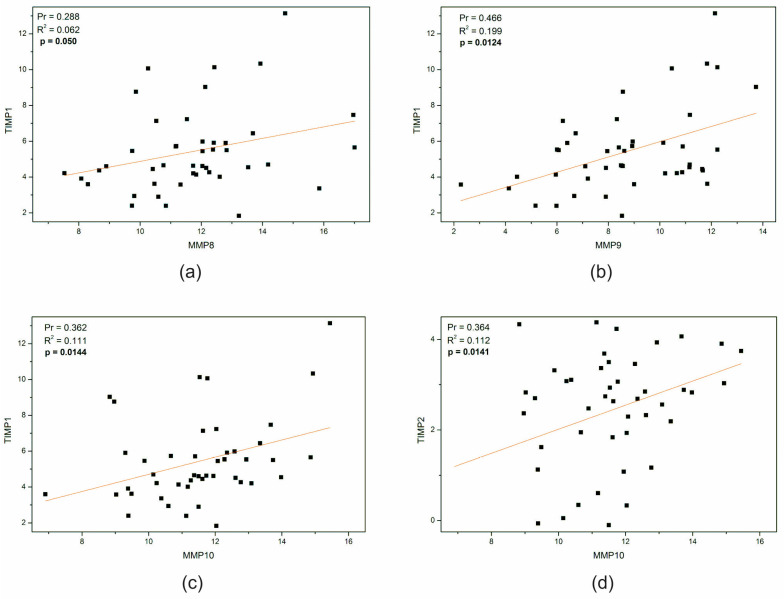
Significant correlation between MMP and TIMP gene expression in all brain tumor samples. Scatter plots with red regression line denote a positive (Pr > 0, increasing slope of regression line) linear correlation between the gene expression of TIMPs and MMPs in all brain tumor samples (OriginPro 8.5 Software). Pearson’s (two-tailed) correlation was used for estimating *p*-values between *MMP8* expression and *TIMP1* (**a**), *MMP9* and *TIMP1* (**b**), *MMP10* and *TIMP1* (**c**), or *TIMP2* (**d**), respectively. Each square represents one single sample expressed as the ratio of the log_10_ 2^−ΔCt^. MMP—matrix metalloproteinase, TIMP—tissue inhibitor of metalloproteinases, Pr—Pearson’s r, R^2^—coefficient of determination, R squared.

**Figure 4 ijms-25-02858-f004:**
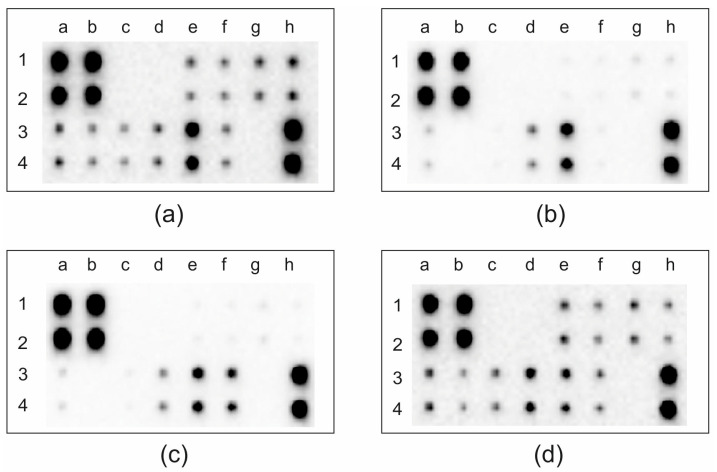
Representative MMP antibody array membranes. (**a**) Control samples; (**b**) meningioma patients; (**c**) astrocytoma patients; (**d**) glioblastoma patients. Proteins were analyzed in duplicate, and their distribution was as follows: MMP1—e1,2; MMP2—f1,2; MMP3—g1,2; MMP8—h1,2; MMP9—a3,4; MMP10—b3,4; MMP13—c3,4; TIMP1—d3,4; TIMP2—e3,4; TIMP4—f3,4. Positive control—a,b(1,2) and h3,4; negative control—c,d(1,2) and g3,4.

**Figure 5 ijms-25-02858-f005:**
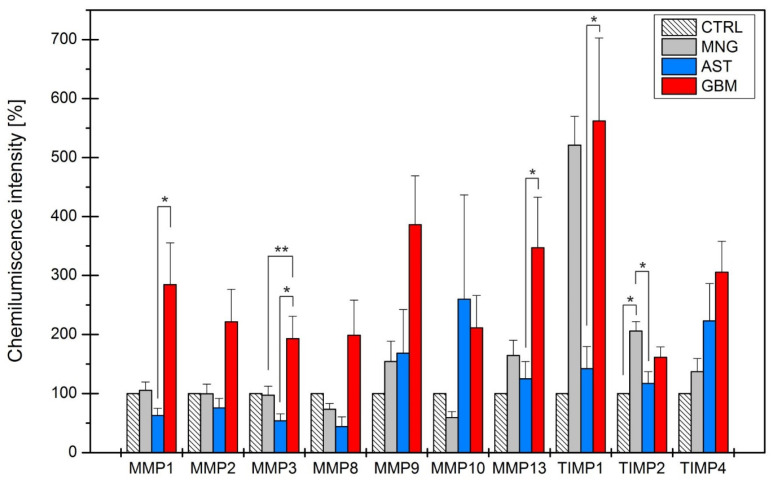
The protein expression of selected proteins in glioblastoma, astrocytoma, and meningioma samples. Each column represents the absolute amount of selected proteins in the glioblastoma (GBM, red), astrocytoma (AST, blue), meningioma (MNG, grey) and control (CTRL, diagonal pattern) groups, where each subgroup stands for an individual gene. One-way ANOVA (OriginPro 8.5 Software) was first carried out to test for differences among all experimental groups. Additionally, Tukey’s test was used to determine the differences between individual groups. The values represented the means ± S.D. of at least three independent experiments; * *p* < 0.05, ** *p* < 0.01. MMP—matrix metalloproteinase, TIMP—tissue inhibitor of metalloproteinases.

**Table 1 ijms-25-02858-t001:** List of selected genes and configuration of RT^2^ PCR Array. Additional information about amplification products’ size and primers is summarized in the previously publication, Dibdiakova et al., 2022 [15].

Category of Genes	Symbol	Protein Product
Metalloproteinases	*MMP1*	Matrix metalloproteinase 1 (interstitial collagenase)
	*MMP2*	Matrix metalloproteinase 2 (gelatinase A)
	*MMP3*	Matrix metalloproteinase 3 (stromelysin 1)
	*MMP8*	Matrix metalloproteinase 8 (neutrophil collagenase)
	*MMP9*	Matrix metalloproteinase 9 (gelatinase B)
	*MMP10*	Matrix metalloproteinase 10 (stromelysin 2)
	*MMP13*	Matrix metalloproteinase 13 (collagenase 3)
Inhibitors of metalloproteinases	*TIMP1*	Tissue inhibitor of metalloproteinases 1
	*TIMP2*	Tissue inhibitor of metalloproteinases 2
	*TIMP4*	Tissue inhibitor of metalloproteinases 4
Housekeeping genes (HKGs)	*ACTB*	β-actin
	*B2M*	β-2-microglobulin
	*GADPH*	Glyceraldehyde-3-phosphate dehydrogenase

## Data Availability

Data is contained within the article.

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
