# Peer review of "Relationship between the Expression of Matrix Metalloproteinases and Their Tissue Inhibitors in Patients with Brain Tumors"

_ijms, 2024, doi:10.3390/ijms25052858_

Round 1

Reviewer 1 Report

Comments and Suggestions for Authors

This paper analyzed samples from brain tumor patients, aiming to seek if there is any difference in matrix metalloproteinases (MMPs) and their inhibitors(TIMPs)among different type of tumors and healthy controls.

Overall the paper was written well, and one of the biggest advantages is the study design. I only have a question: what is the tumor grade, and how that would affect the analysis if the data is further stratified by grade?

I'm not a fan of correlation analysis, either. More of the mechanism/background support would need for that part. 

Author Response

Thank you very much for taking the time to review this manuscript. Please find the detailed responses below and the corresponding revisions in track changes in the re-submitted files.

Response 1: Very good point and thanks for the heads up at the same time. This issue is addressed in Figure 2b (due to the larger group only for mRNA samples), however, where we forgot to categorize the degree of malignancy. Therefore, we have added the required information in the “4.1. Tissue samples” (line 354) subsection and also directly below the mentioned figure (line 148):

“In addition, we divided the patient samples into two groups into benign and malignant. The benign group consisted of meningioma patients (n = 19) and two astrocytoma patients with a maximum tumor grade of II. In the malignant group, glioblastoma patients (n = 20) with grade IV predominated, followed by seven cases of grade III astrocytomas.”

“(b) Fold regulation of MMPs and TIMPs genes between malignant (n = 27) and benign (n = 21) samples.”

Response 2: We assume that you are reflecting on Figure 3, where we used correlation analysis. We have decided to provide a description of the analysis only below the figure. The analysis itself is a two-stage solution where Pearson correlation is added to the classical linear regression. The Pearson correlation measures the strength and direction between two numeric variables while simple linear regression describes the linear relationship between a response variable and an explanatory variable. We must also add that we conducted all pairwise correlations between MMPs and TIMPs, but we depicted only those that turned out to be significantly important. If you require, we can describe the correlation in the methodology section, but for now we are just adding information about the software used under the figure line 176):

“Scatter plots with red regression line denote positive (Pr > 0, increasing slope of regression line) linear correlation between gene expression of TIMPs and MMPs in all brain tumor samples (OriginPro 8.5 Software).”

Reviewer 2 Report

Comments and Suggestions for Authors

In presented paper authors  described well conducted project touching imbalance between MMPs/TIMPs homeostasis in intracerebral tumors tissues. But:

- In "Introduction"  line 58 and in "Discussion" line 299  authors evoke the role of  angiogenesis  as key feature   in tumor development .

So,  in methodology on the level  of presented study design  -   Please explain the absence of analysis of classical angiogenic factors and,   than,  their influence  in MMPs/TIMPs interplay in context of cell migration or tumor invasiveness 

- In "Discussion" line 330 - please explain " unclear mechanism of action   [46]"

The information from lines 446 and 447 please provide also in part 4.1 Tissue samples. 

Author Response

Thank you very much for taking the time to review this manuscript. Please find the detailed responses below and the corresponding revisions in track changes in the re-submitted files.

Response 1: Thank you for pointing this out. Our explanation for the absence of classical angiogenic factors is as follows:

-           We acknowledge that it would be interesting to investigate and further characterize changes in the presence of these factors at the genomic or proteomic level; however, our study was not as rigorously focused on monitoring angiogenesis, as we consider it to be a rather large chapter, which it would be appropriate to deal with in more detail.

Regarding their influence in MMPs/TIMPs interplay in context of cell migration or tumor invasiveness:

-           Angiogenesis is known to play a key role in the growth and progression of cancer cells. The formation of new blood vessels is largely influenced by the activity of MMPs and, in particular, the mutual balance between MMPs and their inhibitors TIMPs. Malignant cells actively secrete matrix metalloproteinases (MMPs), initiating a remodeling process within the extracellular matrix (ECM). This restructuring creates a conducive pathway for the emergence of new blood vessels. Importantly, MMPs play a dual role by not only facilitating ECM alterations but also by promoting the release and activation of key angiogenic factors like vascular endothelial growth factor (VEGF) or basic fibroblast growth factor (bFGF).

For the above reasons, we have decided to keep the text in the introduction, while deleting the section in the discussion, line 299: Inverse co-expression of TIMP4 and MMP1 could partly explain contrary findings. Increased MMP1 expression negatively regulates tumorigenicity and angiogenesis. The immunoblot analysis showed that in the absence of MMP1, the protein level of TIMP4 is significantly increased [33]. A similar trend is also visible in the images from our immunoblot analysis.

Response 2: Yes, we agree with this comment, and we attach additional text after that line (331): 

“They summarized that the essence of the role of TIMP2 in cancer development is based on the protein's concentration within individual tissue. A low concentration of TIMP2 activates matrix metalloproteinase-2, whereas a high concentration inhibits their activity. This phenomenon is attributed to the origin of malignant tissue, as well as various factors (stage of diseases and tumor size) or regulatory molecules (topoisomerase II, p53 and bcl-2).”

Response 3: We are sorry, but we are unable to discern which information from line 446 the reviewer is requesting to include in the methodology section.

“These discrepancies underscore the importance of MMPs as potential targets for research aimed at developing personalized therapies for brain tumors. We observed an increase in MMP2 mRNA levels in meningiomas compared to healthy samples, where MMP1 and MMP13 expression was not amplified in control group.”
